# PRUNING DEEP NEURAL NETWORKS FROM A SPARSITY PERSPECTIVE

**Enmao Diao**\*
Department of Electrical
and Computer Engineering
Duke University
Durham, NC 27705, USA
`enmao.diao@duke.edu`

**Ganghua Wang**\*
School of Statistics
University of Minnesota
Minneapolis, MN 55455, USA
`wang9019@umn.edu`

**Jiawei Zhang**
School of Statistics
University of Minnesota
Minneapolis, MN 55455, USA
`zhan4362@umn.edu`

**Yuhong Yang**
School of Statistics
University of Minnesota
Minneapolis,
MN 55455, USA
`yangx374@umn.edu`

**Jie Ding**
School of Statistics
University of Minnesota
Minneapolis,
MN 55455, USA
`dingj@umn.edu`

**Vahid Tarokh**
Department of Electrical
and Computer Engineering
Duke University
Durham, NC 27705, USA
`vahid.tarokh@duke.edu`

## ABSTRACT

In recent years, deep network pruning has attracted significant attention in order to enable the rapid deployment of AI into small devices with computation and memory constraints. Pruning is often achieved by dropping redundant weights, neurons, or layers of a deep network while attempting to retain a comparable test performance. Many deep pruning algorithms have been proposed with impressive empirical success. However, existing approaches lack a quantifiable measure to estimate the compressibility of a sub-network during each pruning iteration and thus may under-prune or over-prune the model. In this work, we propose PQ Index (PQI) to measure the potential compressibility of deep neural networks and use this to develop a Sparsity-informed Adaptive Pruning (SAP) algorithm. Our extensive experiments corroborate the hypothesis that for a generic pruning procedure, PQI decreases first when a large model is being effectively regularized and then increases when its compressibility reaches a limit that appears to correspond to the beginning of underfitting. Subsequently, PQI decreases again when the model collapse and significant deterioration in the performance of the model start to occur. Additionally, our experiments demonstrate that the proposed adaptive pruning algorithm with proper choice of hyper-parameters is superior to the iterative pruning algorithms such as the lottery ticket-based pruning methods, in terms of both compression efficiency and robustness. Our code is available here.

## 1 INTRODUCTION

Over-parameterized deep neural networks have been applied with enormous success in a variety of fields, including computer vision (Krizhevsky et al., 2012; He et al., 2016b; Redmon et al., 2016), natural language processing (Devlin et al., 2018; Radford et al., 2018), audio signal processing (Oord et al., 2016; Schneider et al., 2019; Wang et al., 2020), and distributed learning (Konečný et al., 2016; Ding et al., 2022; Diao et al., 2022). These deep neural networks have significantly expanded in size. For example, LeNet-5 (LeCun et al., 1998) (1998; image classification) has 60 thousand parameters whereas GPT-3 (Brown et al., 2020) (2020; language modeling) has 175 billion parameters. This rapid growth in size has necessitated the deployment of a vast amount of computation, storage, and energy resources. Due to hardware constraints, these enormous model sizes may be a barrier to deployment in some edge devices such as mobile phones and virtual assistants. This has greatly increased interest in deep neural network compression/pruning. To this end, various researchers have developed empirical methods of building much simpler networks with similar performance based

---

\*Equally contributed

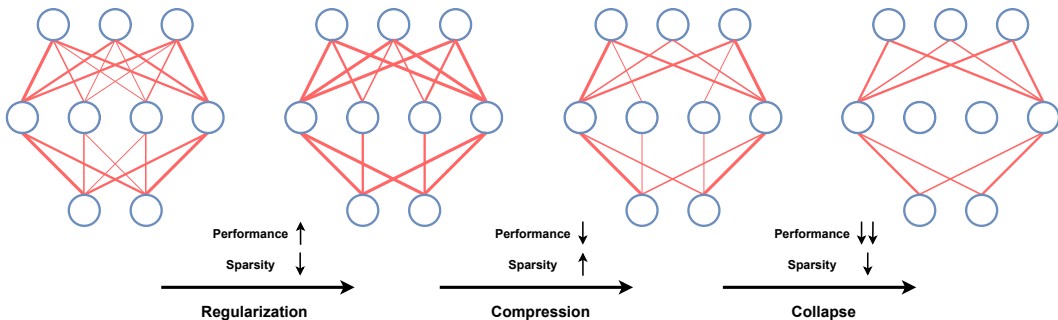

Figure 1: An illustration of our hypothesis on the relationship between sparsity and compressibility of neural networks. The width of connections denotes the magnitude of model parameters.

on pre-trained networks (Han et al., 2015; Frankle & Carbin, 2018). For example, Han et al. (2015) demonstrated that AlexNet (Krizhevsky et al., 2017) could be compressed to retain only $3\%$ of the original parameters on the ImageNet dataset without impacting classification accuracy.

An important topic of interest is the determination of limits of network pruning. An overly pruned model may not have enough expressivity for the underlying task, which may lead to significant performance deterioration (Ding et al., 2018). Existing methods generally monitor the prediction performance on a validation dataset and terminate pruning when the performance falls below a pre-specified threshold. Nevertheless, a quantifiable measure for estimating the compressibility of a sub-network during each pruning iteration is desired. Such quantification of compressibility can lead to the discovery of the most parsimonious sub-networks without performance degradation.

In this work, we connect the compressibility and performance of a neural network to its sparsity. In a highly over-parameterized network, one popular assumption is that the relatively small weights are considered redundant or non-influential and may be pruned without impacting the performance. Let us consider the sparsity of a non-negative vector $w = [w_1, \ldots, w_d]$, since sparsity is related only to the magnitudes of entries. Suppose $S(w)$ is a sparsity measure, and a larger value indicates higher sparsity. Hurley & Rickard (2009) summarize six properties that an ideal sparsity measure should have, originally proposed in economics (Dalton, 1920; Rickard & Fallon, 2004). They are

(D1) Robin Hood. For any $w_i > w_j$ and $\alpha \in (0, (w_i - w_j)/2)$, we have $S([w_1, \ldots, w_i - \alpha, \ldots, w_j + \alpha, \ldots, w_d]) < S(w)$.

(D2) Scaling. $S(\alpha w) = S(w)$ for any $\alpha > 0$.

(D3) Rising Tide. $S(w + \alpha) < S(w)$ for any $\alpha > 0$ and $w_i$ not all the same.

(D4) Cloning. $S(w) = S([w, w])$.

(P1) Bill Gates. For any $i = 1, \ldots, d$, there exists $\beta_i > 0$ such that for any $\alpha > 0$ we have

$$S([w_1, \ldots, w_i + \beta_i + \alpha, \ldots, w_d]) > S([w_1, \ldots, w_i + \beta_i, \ldots, w_d]).$$

(P2) Babies. $S([w_1, \ldots, w_d, 0]) > S(w)$ for any non-zero $w$.

Hurley & Rickard (2009) point out that only Gini index satisfies all six criteria among a comprehensive list of sparsity measures. In this work, we propose a measure of sparsity named PQ Index (PQI). To the best of our knowledge, PQI is the first measure related to the norm of a vector that satisfies all the six properties above. Therefore, PQI is an ideal indicator of vector sparsity and is of its own interest. We suggest using PQI to infer the compressibility of neural networks. Furthermore, we discover the relationship between the performance and sparsity of iteratively pruned models as illustrated in Figure 1. Our hypothesis is that *for a generic pruning procedure, the sparsity will first decrease when a large model is being effectively regularized, then increase when its compressibility reaches a limit that corresponds to the start of underfitting, and finally decrease when the model collapse occurs, i.e., the model performance significantly deteriorates.*

Our intuition is that the pruning will first remove redundant parameters. As a result, the sparsity of model parameters will decrease and the performance may be improved due to regularization. When the model is further compressed, part of the model parameters will become smaller when the

model converges. Thus, the sparsity will increase and the performance will moderately decrease. Finally, when the model collapse starts to occur, all attenuated parameters are removed and the remaining parameters become crucial to maintain the performance. Therefore, the sparsity will decrease and performance will significantly deteriorate. Our extensive experiments on pruning algorithms corroborate the hypothesis. Consequently, PQI can infer whether a model is inherently compressible. Motivated by this discovery, we also propose the Sparsity-informed Adaptive Pruning (SAP) algorithm, which can compress more efficiently and robustly compared with iterative pruning algorithms such as the lottery ticket-based pruning methods. Overall, our work presents a new understanding of the inherent structures of deep neural networks for model compression. Our main contributions are summarized below.

1. We propose a new notion of sparsity for vectors named PQ Index (PQI), with a larger value indicating higher sparsity. We prove that PQI meets all six properties proposed by (Dalton, 1920; Rickard & Fallon, 2004), which capture the principles a sparsity measure should obey. Among 15 commonly used sparsity measures, the only other measure satisfying all properties is Gini Index (Hurley & Rickard, 2009). Thus, norm-based PQI is an ideal sparsity/equity measure and may be of independent interest to many areas, e.g., signal processing and economics.

2. We develop a new perspective on the compressibility of neural networks. In particular, we measure the sparsity of pruned models by PQI and postulate the above hypothesis on the relationship between sparsity and compressibility of neural networks.

3. Motivated by our proposed PQI and hypothesis, we further develop a Sparsity-informed Adaptive Pruning (SAP) algorithm that uses PQI to choose the pruning ratio adaptively. In particular, the pruning ratio at each iteration is decided based on a PQI-related inequality. In contrast, Gini Index does not have such implications for the pruning ratio.

4. We conduct extensive experiments to measure the sparsity of pruned models and corroborate our hypothesis. Our experimental results also demonstrate that SAP with proper choice of hyper-parameters can compress more efficiently and robustly compared with iterative pruning algorithms such as the lottery ticket-based pruning methods.

## 2 RELATED WORK

**Model compression** The goal of model compression is to find a smaller model that has comparable performance to the original model. A smaller model saves storage and computation resources, boosts training, and facilitates the deployment of the model to devices with limited capacities, such as mobile phones and virtual assistants. Therefore, model compression is vital for deploying deep neural networks with millions or even billions of parameters. Various model compression methods have been proposed for neural networks. Among them, pruning is one of the most popular and effective approach (LeCun et al., 1989; Hagiwara, 1993; Han et al., 2015; Hu et al., 2016; Luo et al., 2017; Frankle & Carbin, 2018; Lee et al., 2018; He et al., 2017). The idea of pruning is the sparsity assumption that many redundant or non-influential neuron connections exist in an over-parameterized model. Thus, we can remove those connections (e.g., weights, neurons, or neuron-like structures such as layers) without sacrificing much test accuracy. Two critical components of pruning algorithms are a pruning criterion that decides which connection to be pruned and a stop criterion that determines when to stop pruning and thus prevent underfitting and model collapse. There are many pruning criteria motivated by different interpretations of redundancy. A widely-used criterion removes the parameters with the smallest magnitudes, assuming that they are less important (Hagiwara, 1993; Han et al., 2015). Besides magnitude-based pruning, one may prune the parameters based on their sensitivity or contribution to the network output (LeCun et al., 1989; Lee et al., 2018; Hu et al., 2016; Soltani et al., 2021) or restrict different model components to share a large proportion of neural weights (Diao et al., 2019; 2021). As for the stop criterion, the common choice is validation: to stop pruning once the test accuracy on a validation dataset falls below a given threshold.

While pruning is a post-processing method that requires a pre-trained model, there are also pre-processing and in-processing methods based on the sparsity assumption. For example, one can add explicit sparse constraints on the network, such as forcing the parameters to have a low-rank structure and sharing weights. Alternatively, one can implicitly force the trained model to be sparse, such as adding a sparsity penalty (e.g., $\ell_1$-norm) to the parameters. In contrast to those sparsity-based compression methods, which find a sub-network of the original one, researchers have also proposed

compressing the model by finding a smaller model with a different architecture. The efforts include knowledge distillation (Hinton et al., 2015) and architecture search (Mushtaq et al., 2021). We refer the reader to (Hoefler et al., 2021) for a comprehensive survey of model compression.

**Theory of model compression** In practice, the compressibility of a model depends on the network architecture and learning task. The pruning usually involves a lot of ad hoc hyper-parameter fine-tuning. Thus, an understanding of model compressibility is urgently needed. There have been some recent works to show the existence of or find a sub-network with guaranteed performance. Arora et al. (2018) show that a model is more compressible if it is more stable to the noisy inputs and provides a generalization error bound of the pruned model. Yang et al. (2022) propose a backward pruning algorithm inspired by approximating functions using $\ell_q$-norm (Wang et al., 2014), and quantify its generalization error. Baykal et al. (2018); Mussay et al. (2019) utilize the concept of coreset to prove the existence of a pruned network with similar performance. The main idea is to sample the parameters based on their importance, and thus selected parameters could preserve the output of the original network. Ye et al. (2020) propose a greedy selection algorithm to reconstruct a network and bound the generalization error for two-layer neural networks. Our work develops a new perspective on the compressibility of neural networks by directly measuring the sparsity of pruned models to reveal the relationship between the sparsity and compressibility of neural networks.

**Sparsity measure** Sparsity is a crucial concept in many fundamental fields such as statistics and signal processing (Tibshirani, 1996; Donoho, 2006; Akçakaya & Tarokh, 2008). Intuitively, sparsity means that the most energy is concentrated in a few elements. For example, a widely-used assumption in high-dimensional machine learning is that the model has an underlying sparse representation. Various sparsity measures have been proposed in the literature from different angles. One kind of sparsity measure originates from sociology and economics. For example, the well-known Gini Index (Gini, 1912) can measure the inequality in a population's wealth or welfare distribution. A highly wealth-concentrated population forms a sparse vector if the vector consists of the wealth of each person. In addition, to measure the diversity in a group, entropy-based measures like Shannon entropy and Gaussian entropy are often used (Jost, 2006). Another kind of sparsity measure has been studied in mathematics and engineering for a long time. A classic measure is the hard sparsity, also known as $\ell_0$-norm, which is the number of non-zero elements in $w$. A small hard sparsity implies that only a few vector elements are active or effective. However, a slight change in the zero-valued element may cause a significant increase in the hard sparsity, which can be undesirable. Thus, its relaxations such as $\ell_p$-norm ($0 < p \leq 1$) are also widely used. For example, $\ell_1$-norm-based constraints or penalties are used for function approximation (Barron, 1993), model regularization and variable selection (Tibshirani, 1996; Chen et al., 2001). Our work proposes the first measure of sparsity related to vector norms that satisfies all the properties shared by the Gini Index (Hurley & Rickard, 2009) and an adaptive pruning algorithm based on our proposed measure of sparsity.

## 3 Pruning with PQ Index

### 3.1 PQ Index

We will prove all the six properties (D1)-(D4) and (P1), (P2), which are mentioned in the introduction, hold for our proposed PQ Index (PQI).

**Definition 1** (PQ Index). *For any $0 < p < q$, the PQ Index of a non-zero vector $w \in \mathbb{R}^d$ is*

$$\mathrm{I}_{p,q}(w) = 1 - d^{\frac{1}{q} - \frac{1}{p}} \frac{\|w\|_p}{\|w\|_q}, \tag{1}$$

*where $\|w\|_p = (\sum_{i=1}^d |w_i|^p)^{1/p}$ is the $\ell_p$-norm of $w$ for any $p > 0$. For simplicity, we will use $\mathrm{I}(w)$ and drop the dependency on $p$ and $q$ when the context is clear.*

**Theorem 1.** *We have $0 \leq \mathrm{I}_{p,q}(w) \leq 1 - d^{\frac{1}{q} - \frac{1}{p}}$, and a larger $\mathrm{I}_{p,q}(w)$ indicates a sparser vector. Furthermore, $\mathrm{I}_{p,q}(w)$ satisfies all the six properties (D1)-(D4) and (P1), (P2).*

**Remark 1** (Sanity check). *For the densest or most equal situation, we have $w_i = c$ for $i = 1, \ldots, d$, where $c$ is a non-zero constant. It can be verified that $\mathrm{I}_{p,q}(w) = 0$. In contrast, the sparsest or most unequal case is that $w_i$'s are all zeros except one of them, and corresponding $\mathrm{I}_{p,q}(w) = 1 - d^{\frac{1}{q} - \frac{1}{p}}$. Note that $\mathrm{I}(w)$ for an all-zero vector is not defined. From the perspective of the number of important elements, an all-zero vector is sparse; however, it is dense from the aspect of energy distribution.*

**Remark 2** (Insights). *The form of* $\mathrm{I}_{p,q}$ *is not a random thought but inherently driven by properties (D1)-(D4). Why do we need the ratio of two norms? It is essentially decided by the requirement of (D2) Scaling. If $S(w)$ involves only a single norm, then $S(w)$ is not scale-invariant. However, since $\ell_r$-norm is homogeneous for all $r > 0$, the ratio of two norms is inherently scale-invariant. Why is there an additional scaling constant $d^{\frac{1}{q} - \frac{1}{p}}$? This is necessary to satisfy (D4) Cloning. Inspired by the well-known Root Mean Squared Error (RMSE), we found out that the additional scaling constant is the correct term to help $\mathrm{I}_{p,q}$ be independent of the vector length. It is essentially appealing for comparing the sparsity of neural networks with different model parameters. Why do we require $p < q$? We find it plays a central role in meeting (D1) and (D3). The insight is that $\|w\|_p$ decreases faster than $\|w\|_q$ when a vector becomes sparser, thus guaranteeing a larger PQ Index.*

**Theorem 2** (PQI-bound on pruning). *Let $M_r$ denote the set of $r$ indices of $w$ with the largest magnitudes, and $\eta_r$ be the smallest value such that $\sum_{i \notin M_r} |w_i|^p \leq \eta_r \sum_{i \in M_r} |w_i|^p$. Then, we have*

$$r \geq d(1 + \eta_r)^{-q/(q-p)}[1 - \mathrm{I}(w)]^{\frac{qp}{q-p}}. \tag{2}$$

**Remark 3.** *The PQI-bound is inspired by Yang et al. (2022) that proposed to use $\|w\|_1/\|w\|_q, q \in (0, 1)$ as a measure of sparsity. We use similar techniques to derive the bound based on our proposed PQ Index. It is worth mentioning that $\|w\|_1/\|w\|_q$ does not satisfy properties (D2), (D4), (P1), and (P4), which an ideal sparsity measure should have Hurley & Rickard (2009). Consequently, we cannot use it to compare the sparsity of models of different sizes. The merit of the PQI-bound is that it applies to iterative pruning of models that involve different numbers of model parameters.*

Recall that the pruning is based on the assumption that parameters with small magnitudes are removable. Therefore, suppose we know $\mathrm{I}(w)$ and $\eta_r$, then we immediately have a lower bound for the retaining ratio of the pruning from Theorem 2. Thus, we can adaptively choose the pruning ratio based on $\mathrm{I}(w)$ and $\eta_r$, which inspires our Sparsity-informed Adaptive Pruning (SAP) algorithm in the following subsection. In practice, $\eta_r$ is unavailable before we decide the pruning ratio. Therefore, we treat it as a hyper-parameter in our experiments. Since $\eta_r$ is non-increasing with respect to $r$, a larger $\eta_r$ means that we assume the model is more compressible and leads to a higher pruning ratio. Experiments show that it is safe to choose $\eta_r = 0$.

### 3.2 SPARSITY-INFORMED ADAPTIVE PRUNING

In this section, we introduce the Sparsity-informed Adaptive Pruning (SAP) algorithm as illustrated in Algorithm 1. Our algorithm is based on the well-known lottery ticket pruning method (Frankle & Carbin, 2018). The lottery ticket pruning algorithm proposes to prune and retrain the model iteratively. Compared with one shot pruning algorithm, which does not retrain the model at each pruning iteration, the lottery ticket pruning algorithm produces pruned models of better performance with the same percent of remaining model parameters. However, both methods use a fixed pruning ratio $P$ at each pruning iteration. As a result, they may under-prune or over-prune the model at earlier or later pruning iterations, respectively. The under-pruned models with spare compressibility require more pruning iterations and computation resources to obtain the smallest neural networks with satisfactory performance. The over-pruned model suffers from underfitting due to insufficient compressibility to maintain the desired performance. Therefore, we propose SAP to adaptively determine the number of pruned parameters at each iteration based on the PQI-bound derived in Formula 2. Furthermore, we introduce two additional hyper-parameters, the scaling factor $\gamma$ and the maximum pruning ratio $\beta$, to make our algorithm further flexible and applicable.

Next, we walk through our SAP algorithm. Before the pruning starts, we randomly generate model parameters $w_{\mathrm{init}}$. We will use $w_{\mathrm{init}}$ to initialize retained model parameters at each pruning iterations. The lottery ticket pruning algorithm shows that the performance of retraining the subnetworks from $w_{\mathrm{init}}$ is better than from scratch. Then, we initialize the mask $m_0$ with all ones. Suppose we have $T$ number of pruning iterations. For each pruning iteration $t = 0, 1, 2, \ldots T$, we first initialize the model parameters $\tilde{w}_t$ and compute the number of model parameters $d_t$ as follows

$$\tilde{w}_t = w_{\mathrm{init}} \odot m_t, \quad d_t = |m_t|, \tag{3}$$

where $\odot$ is the Hadamard product. After training the model parameters $\tilde{w}_t$ with $m_t$ by freezing the gradient for $E$ epoch, we arrive at trained model parameters $w_t$. Upon this point, our algorithm has no difference from the classical lottery ticket pruning algorithm. The lottery ticket pruning algorithm will then prune $d \cdot P$ parameters from $m_t$ according to the magnitude of $w_t$ and finally create new mask $m_{t+1}$.

---

**Algorithm 1** Sparsity-informed Adaptive Pruning (SAP)

---

**Input:** model parameters $w$, mask $m$, norm $0 < p < q$, compression hyper-parameter $\eta_r$, scaling factor $\gamma$, maximum pruning ratio $\beta$, number of epochs $E$, and number of pruning iterations $T$.

Randomly generate model parameters $w_{\text{init}}$

  Initialize mask $m_0$ with all ones

  **for** each pruning iteration $t = 0, 1, 2, \ldots T$ **do**

    Initialize model parameters $\tilde{w}_t = w_{\text{init}} \odot m_t$

    Compute the number of model parameters $d_t = |m_t|$

    Train the model parameters $\tilde{w}_t$ with $m_t$ for $E$ epochs and arrive at $w_t$

    Compute PQ Index $\mathrm{I}(w_t) = 1 - d_t^{\frac{1}{q} - \frac{1}{p}} \frac{\|w_t\|_p}{\|w_t\|_q}$

    Compute the lower bound of the number of retained model parameters

    $r_t = d_t (1 + \eta_r)^{-q/(q-p)} [1 - \mathrm{I}(w_t)]^{\frac{qp}{q-p}}$

    Compute the number of pruned model parameters

    $c_t = \lfloor d_t \cdot \min(\gamma(1 - \frac{r_t}{d_t}), \beta) \rfloor$

    Prune $c_t$ model parameters with the smallest magnitude based on $w_t$ and $m_t$

    Create new mask $m_{t+1}$

**end**

**Output:** The pruned model parameters $w_T$ and mask $m_T$.

---

After arriving at $w_t$, our proposed SAP will compute the PQ Index, denoted by $\mathrm{I}(w_t)$, and the lower bound of the number of retrained model parameters, denoted by $r_t$, as follows

$$\mathrm{I}(w_t) = 1 - d_t^{\frac{1}{q} - \frac{1}{p}} \frac{\|w_t\|_p}{\|w_t\|_q}, \qquad r_t = d_t (1 + \eta_r)^{-q/(q-p)} [1 - \mathrm{I}(w_t)]^{\frac{qp}{q-p}}. \tag{4}$$

Then, we compute the number of pruned model parameters $c_t = \lfloor d_t \cdot \min(\gamma(1 - \frac{r_t}{d_t}), \beta) \rfloor$. Here, we introduce $\gamma$ to accelerate or decelerate the pruning at initial pruning iterations. Specifically, $\gamma > 1$ or $< 1$ encourages the pruning ratio to be larger or smaller than the pruning ratio derived from the PQI-bound, respectively. As the retraining of pruned models is time-consuming, it is appealing to efficiently obtain the smallest pruned model with satisfactory performance for a small number of pruning iterations. In addition, we introduce the maximum pruning ratio $\beta$ to avoid excessive pruning because we will compress more than the PQI-bound if $\gamma > 1$ and may completely prune all model parameters. If we set $\gamma = 1$, $\beta$ can be safely omitted. In our experiments, we set $\beta = 0.9$ only to provide minimum protection from excessive pruning at each pruning iteration. Finally, we prune $c_t$ model parameters with the smallest magnitude based on $w_t$ and $m_t$, and finally create new mask $m_{t+1}$ for the next pruning iteration.

## 4 EXPERIMENTAL STUDIES

### 4.1 EXPERIMENTAL SETUP

We conduct experiments with FashionMNIST (Xiao et al., 2017), CIFAR10, CIFAR100 (Krizhevsky et al., 2009), and TinyImageNet (Le & Yang, 2015) datasets. Our backbone models are Linear, Multi-Layer Perceptron (MLP), Convolutional Neural Network (CNN), ResNet18, ResNet50 (He et al., 2016a), and Wide ResNet28x8 (WResNet28x8) (Zagoruyko & Komodakis, 2016). We run experiments for $T = 30$ pruning iterations with Linear, MLP, and CNN, and $T = 15$ pruning iterations with ResNet18. We compare the proposed SAP with two baselines, including 'One Shot' and 'Lottery Ticket' (Frankle & Carbin, 2018) pruning algorithms. The difference between 'One Shot' and 'Lottery Ticket' pruning algorithms is that 'One Shot' prunes $d \cdot P$ model parameters at each pruning iteration from $w_0$ instead of $w_t$. We have $P = 0.2$ throughout our experiments. We compare the proposed PQ Index ($p = 0.5$, $q = 1.0$) with the well-known Gini Index (Gini, 1912) to validate its effectiveness in evaluating sparsity. Furthermore, we perform pruning on various pruning scopes, including 'Neuron-wise Pruning,' 'Layer-wise Pruning,' and 'Global Pruning.' In particular, 'Global Pruning' gather all model parameters as a vector for pruning, while 'Neuron-wise Pruning' and 'Layer-wise Pruning' prune each neuron and layer of model parameters separately. The number of neurons at each layer is equal to the output size of that layer. For example, 'One Shot' and 'Lottery Ticket methods with 'Neuron-wise Pruning' prune $d_i \cdot P$ model parameters of each neuron, where $d_i$ refers to the size of each neuron. Similarly, SAP computes the PQ Index and the number of pruned

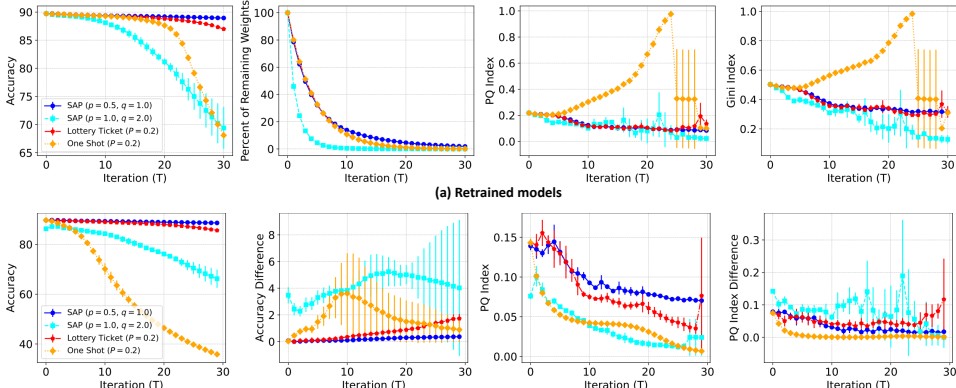

Figure 2: Results of (a) retrained and (b) pruned models at each pruning iteration for 'Global Pruning' with FashionMNIST and MLP, where (a) is obtained from the models after retraining and (b) is directly from those after pruning.

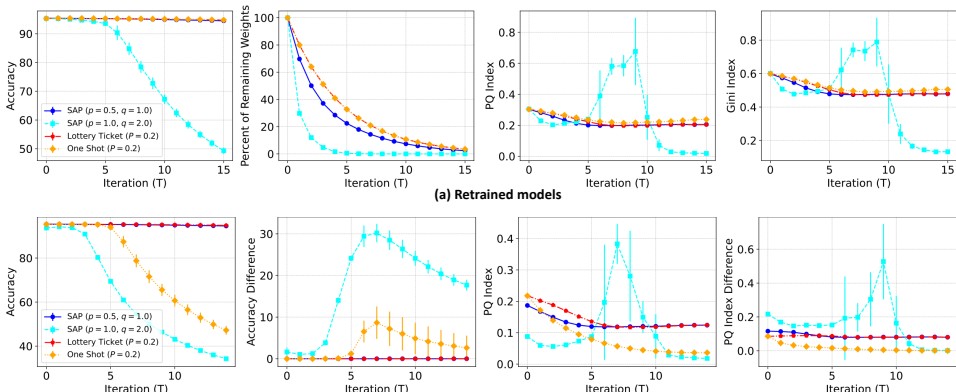

Figure 3: Results of (a) retrained and (b) pruned models at each pruning iteration for 'Global Pruning' with CIFAR10 and ResNet18.

model parameters for neuron $i$. Details of the model architecture and learning hyper-parameters are included in the Appendix. We conducted four random experiments with different seeds, and the standard deviation is shown in the error bar of all figures. Further experimental results can be found in the Appendix.

## 4.2 EXPERIMENTAL RESULTS

**Retrained models** We demonstrate the results of retrained models ($w_t \odot m_t$) at each pruning iteration in Figure 2(a) and 3(a). In particular, we illustrate the performance, percent of remaining weights, PQ Index, and Gini Index at each pruning iteration. In these experiments, $\eta_r$ and $\gamma$ are set to 0 and 1 to prevent interference. The 'Lottery Ticket' method outperforms 'One Shot' as expected. SAP ($p = 1.0, q = 2.0$) compresses more aggressively than SAP ($p = 0.5, q = 1.0$). Recall that SAP adaptively adjusts the pruning ratio based on the sparsity of models, while 'One Shot' and 'Lottery Ticket' have a fixed pruning ratio at each pruning iteration. In Figure 2(a), SAP ($p = 1.0, q = 2.0$) at around $T = 10$ achieves similar performance as 'Lottery Ticket' at around $T = 25$. Furthermore, SAP ($p = 0.5, q = 1.0$) and 'Lottery Ticket' have similar pruning ratio before $T = 10$, but SAP ($p = 0.5, q = 1.0$) adaptively lowers the pruning ratio and thus prevents the performance from deteriorating like 'Lottery Ticket' does after $T = 25$. In Figure 3(a), we observe similar fast pruning phenomenon of SAP ($p = 1.0, q = 2.0$). Meanwhile, SAP ($p = 0.5, q = 1.0$) performs similar to 'Lottery Ticket' but prunes more than 'Lottery Ticket' does. Consequently, by carefully choosing $p$ and $q$, SAP can provide more efficient and robust pruning. As for the sparsity of retrained models, the result of the PQ Index is aligned with the Gini Index, which experimentally demonstrates that the PQ Index can effectively measure the sparsity of model parameters. Furthermore, the dynamics of the sparsity also corroborates our hypothesis, as shown by 'One Shot' in Figure 2(a) and SAP

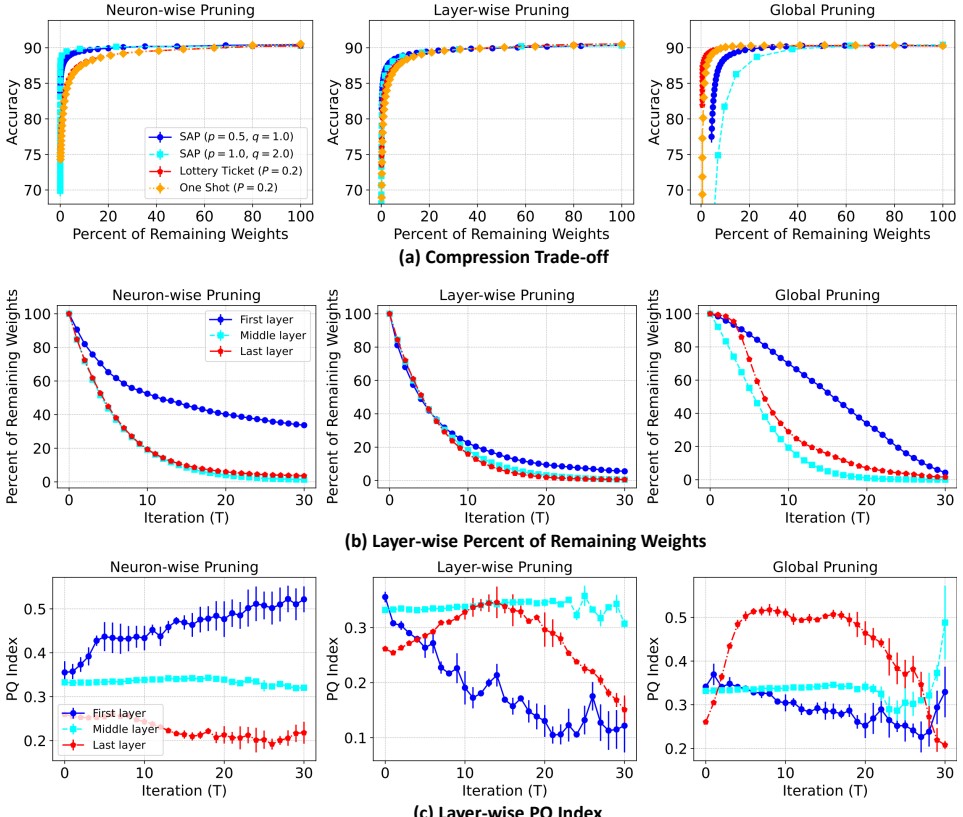

Figure 4: Results of various pruning scopes regarding (a) compression trade-off, (b) layer-wise percent of remaining weights, and (c) layer-wise PQ Index for CIFAR10 and CNN. (b, c) are performed with SAP ($p = 0.5$, $q = 1.0$).

($p = 1.0$, $q = 2.0$) in Figure 3(a). The results show that an ideal pruning procedure should avoid a rapid increase in sparsity.

**Pruned models** We demonstrate the results of pruned models at each pruning iteration in Figure 2(b) and 3(b). In particular, we illustrate the performance, performance difference, PQ Index, and PQ Index difference at each pruning iteration. The performance and PQ Index are computed directly from the pruned models without retraining. The performance and PQ Index difference is between the retrained ($w_0 \odot m_t$ for 'One Shot' and $w_t \odot m_t$ for 'Lottery Ticket' and SAP) and pruned models ($w_0 \odot m_{t+1}$ for 'One Shot' and $w_t \odot m_{t+1}$ for 'Lottery Ticket' and SAP). The results of performance difference show that the performance of pruned models without retraining from SAP can perform close to that of retrained models. Furthermore, the sparsity of pruned models shows that iterative pruning generally decreases the sparsity of pruned models. Meanwhile, the sparsity difference of pruned models provides a sanity check by showing that pruning at each pruning iteration decreases the sparsity of retrained models.

**Pruning scopes** We demonstrate various pruning scopes regarding compression trade-off, layer-wise percent of remaining weights, and layer-wise PQ Index in Figure 4. The results of the compression trade-off show that SAP with 'Global Pruning' may perform worse than 'One Shot' and 'Lottery Ticket' when the percent of remaining weights is small. As illustrated in the 'Global Pruning' of Figure 4(b), the first layer has not been pruned enough. It is because SAP with 'Global Pruning' measures the sparsity of all model parameters in a vector, and the magnitude of the parameters of the first layer and other layers may not be at the same scale. As a result, the parameters of the first layer will not be pruned until late pruning iterations. However, SAP with 'Neuron-wise Pruning' and 'Layer-wise Pruning' perform better than 'One Shot' and 'Lottery Ticket.' SAP can adaptively adjust the pruning ratio of each neuron and layer, but 'One Shot' and 'Lottery Ticket' may over-prune specific neurons and layers because they adopt a fixed pruning ratio. Interestingly, the parameters of the first layer are pruned more aggressively by 'Neuron-wise Pruning' than 'Global Pruning' in the

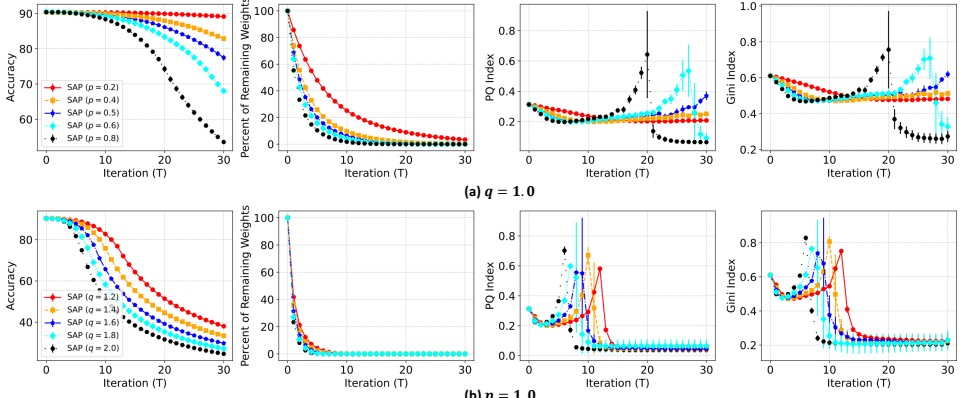

Figure 5: Ablation studies of $p$ and $q$ for global pruning with CIFAR10 and CNN.

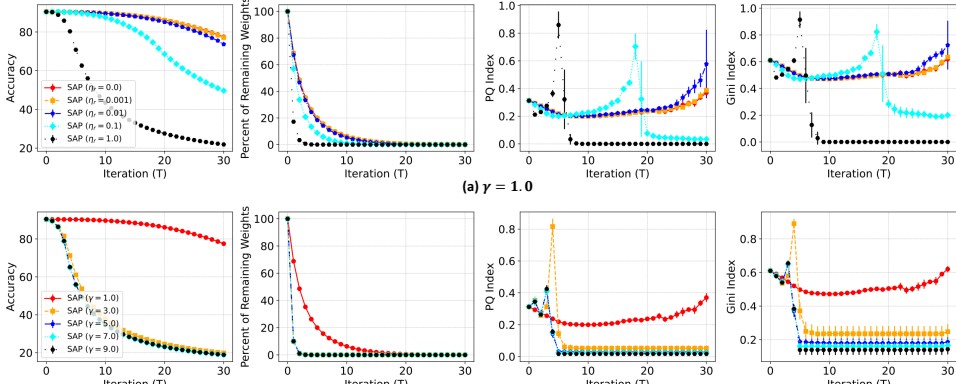

Figure 6: Ablation studies of $\eta_r$ and $\gamma$ for global pruning with CIFAR10 and CNN.

early pruning iterations. However, they are not pruned by 'Neuron-wise Pruning' in the late pruning iterations, while 'Global Pruning' still prunes them aggressively. It aligns with the intuition that the initial layers of CNN are more important to maintain the performance, e.g., Gale et al. (2019) observed that the first layer was often more important to model quality and pruned less than other layers. Furthermore, the PQ Index of 'Neuron-wise Pruning' is also more stable than the other two pruning scopes, which indicates that 'Neuron-wise Pruning' is more appropriate for SAP, as the PQ Index is computed more precisely.

**Ablation studies** We demonstrate ablation studies of $p$ and $q$ in Figure 5. In Figure 5(a), we fix $q = 1.0$ and study the effect of $p$. The results show that SAP prunes more aggressively when $q = 1.0$ and $p$ is close to $q$. In Figure 5(b), we fix $p = 1.0$ and study the effect of $q$. The results show that SAP prunes more aggressively when $p = 1.0$ and $q$ is distant from $p$. We demonstrate ablation studies of $\eta_r$ and $\gamma$ in Figure 6. In Figure 6(a), we fix $\gamma = 1.0$ and study the effect of $\eta_r$. The results show that SAP prunes more aggressively when $\eta_r > 0$. In Figure 6(b), we fix $\eta_r = 0.0$ and study the effect of $\gamma$. The results show that SAP prunes more aggressively when $\gamma > 1$. Interestingly, the performance of our results roughly follows a logistic decay model due to the adaptive pruning ratio, and the inflection point corresponds to the peak of the sparsity measure. Moreover, the dynamics of the sparsity measure of SAP with various ablation studies also corroborate our hypothesis.

## 5 CONCLUSION

We proposed a new notion of sparsity for vectors named PQ Index (PQI), which follows the principles a sparsity measure should obey. We develop a new perspective on the compressibility of neural networks by measuring the sparsity of pruned models. We postulate a hypothesis on the relationship between the sparsity and compressibility of neural networks. Motivated by our proposed PQI and hypothesis, we further develop a Sparsity-informed Adaptive Pruning (SAP) algorithm that uses PQI to choose the pruning ratio adaptively. Our experimental results demonstrate that SAP can compress more efficiently and robustly than state-of-the-art algorithms.

## ACKNOWLEDGMENTS

This work was supported in part by the Office of Naval Research (ONR) under grant number N00014-21-1-2590.

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
