# OpenReview forum: "Pruning Deep Neural Networks from a Sparsity Perspective"
_ICLR.cc/2023/Conference — ICLR 2023 poster_

### Official Review · Reviewer_UD4g · 2022-10-25

**Confidence:** 4
**Correctness:** 2
**Technical Novelty And Significance:** 3
**Empirical Novelty And Significance:** 2
**Recommendation:** 6

**Clarity, Quality, Novelty And Reproducibility:**

Overall speaking, the paper is well-written and easy to follow. The idea of this paper is novel. The reproducibility should be good since no large datasets and models are included.

**Strength And Weaknesses:**

+The authors developed the PQ index, which satisfies the properties of an ideal sparsity measure.

+They further proposed the sparsity-informed adaptive pruning method based on the PQ index.

-It's unclear how SAP can reach a pre-defined pruning rate given some iterations since SAP is an adaptive pruning algorithm. Many hyperparameters are related to the pruning rate per iteration, like $\eta_r$, $\beta$, and $\gamma$.

-Authors mentioned that "pruning is based on the assumption that parameters with small magnitudes are removable." However, this is not always the case, many methods use other criteria to measure the importance of each weight besides its magnitude.

-The paper didn't include large-scale datasets and models. As a result, the scalability of this method is not clear. Large models like ResNet-50 on ImageNet should also be added. The initial version of the lottery ticket can not work for large models, and the authors should show that SAP does not have the same problem.

**Summary Of The Paper:**

In this work, the authors proposed PQ Index (PQI) to measure the potential compressibility of deep neural networks and it is used to develop a Sparsity-informed Adaptive Pruning (SAP) algorithm.

**Summary Of The Review:**

This paper proposed a new sparsity measure: PQ index, and they developed SAP for pruning. However, some details of the SAP algorithm are missing, and there are no experiments for large-scale datasets and models.

---

> ### Author Response · Authors · 2022-11-19
> **Response**
>
> Thank you for your time and constructive comments. We have conducted additional experiments, revised our paper, highlighted the changes, and tried our best to address all the comments below. We hope the responses and revisions will be viewed favorably.
>
> 1. It's unclear how SAP can reach a pre-defined pruning rate given some iterations since SAP is an adaptive pruning algorithm. Many hyperparameters are related to the pruning rate per iteration, like $\eta_r$, $\beta$, $\gamma$.
> > The merit of SAP is that it can adaptively choose the pruning ratio so that the model can be appropriately pruned and retain high accuracy. A pre-defined pruning rate may over- or under-prune the initial model. $\eta_r$, $\gamma$, and $\beta$ are hyper-parameters to scale the pruning ratio obtained from PQI (Theorem 2). They are introduced for better performance and are not necessary. In particular, suppose $r$ is the number of weights we want to preserve in each pruning iteration, and it is unclear how we should set it in practice. It varies in each iteration and for different datasets and models. In contrast, Theorem 2 provides a theoretical-guided way to choose $r$, which is determined by PQI and $\eta_r$. Here, $\eta_r$ has an interpretation as our expectation of the model sparsity. Moreover, it is safe to set $\eta_r$ to zero from our experiment results. Also, $\gamma$ and $\beta$ could be set to one by default. We conduct extensive ablation studies to demonstrate the effect and insights of those hyper-parameters.
>
> 2. Authors mentioned that "pruning is based on the assumption that parameters with small magnitudes are removable." However, this is not always the case, many methods use other criteria to measure the importance of each weight besides its magnitude.
> > We agree that there are many other criteria to measure the importance of each weight, and have changed our statement in the revision. In our paper, we choose magnitude-based pruning and focus on demonstrating the relationship between the performance of iterative pruning and the dynamics of PQI. Moreover, the proposed PQI develops a better iterative pruning algorithm that can adaptively determine the pruning ratio. Theorem 2 is how we connect PQI with magnitude-based pruning methods. As for the pruning criteria, it is also possible and interesting to integrate PQI to facilitate the pruning procedure. We describe the future work in Appendix A of the revision.
>
> 3. The paper didn't include large-scale datasets and models. As a result, the scalability of this method is not clear. Large models like ResNet-50 on ImageNet should also be added. The initial version of the lottery ticket can not work for large models, and the authors should show that SAP does not have the same problem.
> > In the revision, we conduct experiments with CIFAR100 (WResNet28x2) illustrated in Figure 9, 17, and 25, and TinyImageNet (ResNet50) illustrated in Figure 10, 18, and 26.  Due to time and resource constraints, we use TinyImageNet instead of ImageNet and add CIFAR100 with WResNet28x2. The demonstrated results are consistent with our previous experimental results.

---

> ### Author Response · Authors · 2022-12-08
> **A kind reminder**
>
> Dear Reviewer UD4g,
>
> We would like to thank you again for the time you dedicated to reviewing our paper and your valuable comments. We believe that we have addressed your concerns. Since the end of Discussion Stage 2 is approaching and we have not heard back from you yet, we would appreciate if you kindly let us know of any other concerns you may have, and if we can be of any further assistance in clarifying any issues.
>
> Best wishes,
>
> Authors

---

### Official Review · Reviewer_Tdbq · 2022-10-25

**Confidence:** 4
**Correctness:** 3
**Technical Novelty And Significance:** 4
**Empirical Novelty And Significance:** 3
**Recommendation:** 6

**Clarity, Quality, Novelty And Reproducibility:**

The introduced PQI is novel and the paper is clear and of high quality. I would encourage the authors to release the code to allow reproducibility.

**Strength And Weaknesses:**

## Strengths
1. The problem of identifying sparsity ratios is important and it is valuable for many pruning algorithms. The introduced PQI has desirable properties and shows improved performance when used to adaptively set sparsity ratios while pruning.
2. The paper is well-written and the claims are backed by theory and experiments

## Weaknesses
1. The final algorithm introduces additional hyperparameters (eg., \eta_r, \gamma, \beta) and is not clear about the necessity and if they are necessary raises concerns about the introduced PQI. Specifically if \eta_r is a hyperparameter then how is it different to having r as a hyperparameter? I presume the algorithm is not sensitive to the value of \eta_r too much but it would be good to clarify this.
2. The argument made in fig.1 is handwavy and not clear how it helps the narrative of the paper. To me, it was a distraction while reading the paper and it would be better to introduce PQI as early as possible.

**Summary Of The Paper:**

The paper introduces a pruning quality index (PQI) that satisfies desired properties (according to previous work) which can be used to set sparsity ratios. They use PQI to obtain sparsity ratios in the lottery ticket algorithm and show improved results.

**Summary Of The Review:**

The introduced PQI index is novel and would be valuable to the community. The benefits are demonstrated in the experiments. However, the necessity of additional hyperparameters raises some questions about the practical value.

## Post rebuttal
I thank the authors for the response. As mentioned by reviewer dVA5, the practical implementation of the method is cumbersome due to additional hyperparameters. Nevertheless, the introduced PQI has value and I'm keeping the marginal-accept score.

---

> ### Author Response · Authors · 2022-11-19
> **Response**
>
> Thank you for your time and constructive comments. We have conducted additional experiments, revised our paper, highlighted the changes, and tried our best to address all the comments below. We hope the responses and revisions will be viewed favorably.
>
> 1. The final algorithm introduces additional hyperparameters (eg., $\eta_r$, $\gamma$, $\beta$) and is not clear about the necessity and if they are necessary raises concerns about the introduced PQI. Specifically if $\eta_r$ is a hyperparameter then how is it different to having r as a hyperparameter? I presume the algorithm is not sensitive to the value of $\eta_r$ too much but it would be good to clarify this.
> > $\eta_r$, $\gamma$, and $\beta$ are hyper-parameters to scale the pruning ratio obtained from PQI (Theorem 2). They are introduced for better performance and are not necessary. In particular, $r$ is the number of weights we want to preserve in each pruning iteration, and it is unclear how we should set it in practice. It varies in each iteration and for different datasets and models. In contrast, Theorem 2 provides a theoretical-guided way to choose $r$ determined by PQI and $\eta_r$. Here, $\eta_r$ has an interpretation as our expectation of the model sparsity. Moreover, it is safe to set $\eta_r$ to zero from our experiment results. Also, $\gamma$ and $\beta$ could be set to one by default. We conduct extensive ablation studies to demonstrate the effect and insights of those hyper-parameters.
>
> 2. The argument made in fig.1 is handwavy and not clear how it helps the narrative of the paper. To me, it was a distraction while reading the paper and it would be better to introduce PQI as early as possible.
> > As you suggested, we describe the general notion of sparsity in the introduction section of the revision. The proposed hypothesis is a crucial discovery of how model sparsity relates to compressibility and performance during iterative pruning. We want to point out that a single PQI value does not imply the performance of a network, but the dynamics of PQI during the pruning procedure is a possible indicator. By observing PQI, we can determine when we should stop iterative pruning to prevent significant performance drops. In particular, one may stop pruning once the PQI starts to increase or suddenly drops from the peak value. During iterative pruning, PQI initially decreases due to regularization (the network becomes denser than before, and performance does not decrease). Then PQI starts to increase due to compression (the network becomes sparser than before and performance slightly drops). Finally, PQI suddenly drops from a peak (the network becomes dense again, and the performance significantly drops).
>
> 3. The introduced PQI is novel and the paper is clear and of high quality. I would encourage the authors to release the code to allow reproducibility.
> > Our source codes are provided in the supplementary materials. We will release our source codes officially once our paper gets published.

---

> ### Author Response · Authors · 2022-12-08
> **A kind reminder**
>
> Dear Reviewer Tdbq,
>
> We would like to thank you again for the time you dedicated to reviewing our paper and your valuable comments. We believe that we have addressed your concerns. Since the end of Discussion Stage 2 is approaching and we have not heard back from you yet, we would appreciate if you kindly let us know of any other concerns you may have, and if we can be of any further assistance in clarifying any issues.
>
> Best wishes,
>
> Authors

---

### Official Review · Reviewer_fMpo · 2022-10-25

**Confidence:** 2
**Correctness:** 4
**Technical Novelty And Significance:** 4
**Empirical Novelty And Significance:** 3
**Recommendation:** 8

**Clarity, Quality, Novelty And Reproducibility:**

Clarity
The context, explanation and technical content are well presented.

Quality and Originality
The PQI measure itself is well presented and seems novel, alongside its application to pruning.

**Strength And Weaknesses:**

Strengths
- A well rounded explanation of the various properties of a sparsity measure and how PQI satisfies them.
- The scope of ablation and results presented analyze the given method well.

Weaknesses
- I encourage the authors to put forward the general notion of sparsity that is assumed across the paper (as defined in $S(w)$) early on in the introduction so that reader can follow the ideas put forward in Fig. 1.
- One of the major issues in the context of pruning literatures' results is the use of MNIST, FashionMNIST and CIFAR10 to evaluate the performance of the proposed model. I encourage the authors to further expand the set of dataset-DNN pairs they experiment on in order to incorporate more real-world data and ensure their observations remain consistent.
- From a more practical perspective, could the authors discuss the absolute limit up to which they can push the sparsity limit of various networks? (Since that is the ultimate goal)
- By extension, could the authors discuss difference in performance values and PQI at the extreme end of sparsity (highlight in existing results)?


**Summary Of The Paper:**

The proposed work emphasizes the problem that during pruning there isn't a concrete way to estimate the compressability of a sub-network, which may lead to over- or under-pruning. The PQIndex is proposed to measure this concept and by extension is used to define the Sparsity-informed Adaptive Pruning (SAP) algorithm. Overall, the proposed work measures the trends in sparsity and how it fluctuates during different levels of pruning.

**Summary Of The Review:**

The novelty of the proposed PQI measure and the pruning algorithm are of keen interest. However, the results could be further improved to make it more comparable to state-of-the-art approaches.

---

> ### Author Response · Authors · 2022-11-19
> **Response**
>
> Thank you for your time and constructive comments. We have conducted additional experiments, revised our paper, highlighted the changes, and tried our best to address all the comments below. We hope the responses and revisions will be viewed favorably.
>
> 1.  I encourage the authors to put forward the general notion of sparsity that is assumed across the paper (as defined in $S(w)$) early on in the introduction so that reader can follow the ideas put forward in Fig. 1.
> > As you suggested, we describe the general notion of sparsity in the introduction section of the revision.
>
> 2. One of the major issues in the context of pruning literatures' results is the use of MNIST, FashionMNIST and CIFAR10 to evaluate the performance of the proposed model. I encourage the authors to further expand the set of dataset-DNN pairs they experiment on in order to incorporate more real-world data and ensure their observations remain consistent.
> > In the revision, we conduct experiments with CIFAR100 (WResNet28x2) illustrated in Figure 9, 17, and 25, and TinyImageNet (ResNet50) illustrated in Figure 10, 18, and 26.  Due to time and resource constraints, we use TinyImageNet instead of ImageNet and add CIFAR100 with WResNet28x2. The demonstrated results are consistent with our previous experimental results.
>
> 3. From a more practical perspective, could the authors discuss the absolute limit up to which they can push the sparsity limit of various networks? (Since that is the ultimate goal)
> > By observing PQI, we can determine when we should stop iterative pruning to prevent significant performance drops. In particular, one may stop pruning once the PQI starts to increase or suddenly drops from the peak value. During iterative pruning, PQI initially decreases due to regularization (the network becomes denser than before, and performance does not decrease). Then PQI starts to increase due to compression (the network becomes sparser than before and performance slightly drops). Finally, PQI suddenly drops from a peak (the network becomes dense again, and the performance significantly drops). We want to point out that a single PQI value does not imply the performance of a network, but the dynamics of PQI during the pruning procedure is a possible indicator.
>
> 4. By extension, could the authors discuss difference in performance values and PQI at the extreme end of sparsity (highlight in existing results)?
> > Thank you for pointing it out. In our paper, we focus on demonstrating the relationship between the performance of iterative pruning and the dynamics of PQI. It is interesting to study how the performance and PQI are related to other pruning methods. We describe the future work in Appendix A of the revision.

---

> ### Author Response · Authors · 2022-12-08
> **A kind reminder**
>
> Dear Reviewer fMpo,
>
> We would like to thank you again for the time you dedicated to reviewing our paper and your valuable comments. We believe that we have addressed your concerns. Since the end of Discussion Stage 2 is approaching and we have not heard back from you yet, we would appreciate if you kindly let us know of any other concerns you may have, and if we can be of any further assistance in clarifying any issues.
>
> Best wishes,
>
> Authors

---

> > ### Comment · Reviewer_fMpo · 2022-12-08
> > **Response**
> >
> > I thank the authors for their reminder. As is, I have minimal outstanding concerns.

---

### Official Review · Reviewer_dVA5 · 2022-10-30

**Confidence:** 4
**Correctness:** 2
**Technical Novelty And Significance:** 4
**Empirical Novelty And Significance:** 2
**Recommendation:** 5

**Clarity, Quality, Novelty And Reproducibility:**

To my knowledge, the contribution of connecting sparsity metrics introduced in economics to weight sparsity in neural networks is novel. Also I found the explanation of the proposed pruning method clear. Reading the text one could implement the method easily. My main concerns are about the evaluation of the proposed method.

**Strength And Weaknesses:**

# Strengths
- Proposal of a grounded sparsity metric. As far as I know, this part is novel and the most exciting part of this work. I appreciate the utilization of cross-domain knowledge.

# Weakness
- Experimental results are not clear and convincing. Lottery ticket pruning does as good or better than proposed method in many plots. Do you do hyper-parameter search for different methods? A fair comparison should include different fraction values for lottery style pruning. Similarly when different p/q values are selected one probably needs to pick different hyper-parameters. Also, plots needs to be improved to highlight the differences between methods. Using log on the y axis (Figure-2/3) can help with this I think.
- "sota algorithms such as LT-based pruning" This statement is not correct. See [1], which show rewinding the weights is a bad choice and [2] which shows gradual magnitude pruning often achieves best results. It's fine not to have SOTA results/algorithms but this statement is not correct and needs to be changed. I recommend using another pruning algorithm like gradual magnitude pruning with or without learning rate recycling (see [2]).
- Authors say in the introduction "the sparsity will first decrease when a large model is being effectively regularized", however I haven't seen any evidence for this claim. Have I missed a plot? It would be nice to show this in a regular training run (like resnet in cifar-10 with learning rate decay and 200 epochs of training). Similarly one could think of experiment that show the usefulness of the PQ-Index at predicting pruning rates. Can PQ-Index predict maximum pruning rate over the course of the training? How well this compare to using a fix magnitude threshold?

# Questions
- Looking at Theorem-2, n_r=0 seems to imply removing only zeros. Is that accurate? If so, would be nice to make this clear.
- Do you include pruned connections (zeros) in your PQI calculations?
- Figure:5-a there is a peak on the PQ-Index at T=20. Why is this? With a high score like this I would expect performance not to drop after pruning.

# Suggestions
- When p=0.5 and q=1, sparsity would be equal to PQ-metric if I understand correctly. d=100, one-hot, sparsity=PQ-index=0.99. It would be nice to do experiments on pretrained models at different widths and sparsities (this can be achieved with different regularization coefficients) and show how well PQ-index provide maximum pruning rate without significant performance drop. It is very difficult to make conclusion in the LT-rewinding setting since, pruning is repeated multiple times and it is not clear a decision was good or not.

# Minor
- (Page 8: "It aligns with the intuition..") [2] shows that early layers are often better kept dense.
- It would be nice to compare the final sparsity distributions found by PQI with the ones provided by the ERK distribution [3].
- "pruning is the most popular and effective approach" probably quantization had much more impact on this due to the int8 support on many recent hardware.
- Remark 3 "... does not satisfy the six properties" -> which of the six? I think some of them are satisfied. It would be nice to be more specific here.
- Lottery ticket prunes p% of the remaining weights. So the #weights pruned at each iteration decreases over time.
- Figure:3, it is not clear which methods does better (like many others).

## After Rebuttal
I thank authors for their extensive response. After reading their response and checking the changes indicated with blue; I decided to keep my score as it is. That's being said, I wouldn't oppose acceptance. Overall I find the the metric quite interesting, however experimental evaluation for proposed pruning method needs to be improved. I recommend authors to focus on showing the utility of PQI directly using the same model but different metrics (e.g. measuring exactly how close it gets to the max-prune rate). Similarly I think the fixed-pruning rate needs to be searched-over for the LT baseline (like trying P=0.2,0.1,0.3) just to make sure gains are not due to better rate average rate (first point in weakness section). Also I like to re-iterate switching the log-scale for plots, as it is still very difficult to compare different curves at the moment.

[1] https://arxiv.org/abs/2003.02389
[2] https://arxiv.org/abs/1902.09574
[3] https://arxiv.org/abs/1911.11134

**Summary Of The Paper:**

This paper introduces a norm based metric (PQ-Index) to quantify weight sparsity of a neural network. The metric is designed to satisfy the six properties proposed by earlier research in economy similar to the Gini Index. PQ-Index is then used to decide how much a neural network can be pruned at a given time during the training. Authors compare their method to Lottery-ticket style iterative pruning on CIFAR and Fashion-MNIST.

**Summary Of The Review:**

Overall quite interesting work, my concerns lie in experimental evaluations of the work. More specifically contributions 2 and 4 in introduction are not well supported:
- Relationship between PQI and compressibility is not measured directly. This can be done with one-shot pruning experiments (See Suggestion above).
- "SAP can compress more efficiently than SOTA algorithms" is also not well supported since, Lottery style iterative magnitude pruning is far from being a SOTA algorithm and most plots doesn't show improved performance (or hard to read).

I recommend authors to focus on showing the utility of the metric they developed and understand its strengths and limitations. I think such contribution alone would make a good paper. If proposing a pruning algorithm, experiments/plots/validation needs to be improved/clarified.

---

> ### Author Response · Authors · 2022-11-19
> **Response**
>
> Thank you for your time and constructive comments. We have conducted additional experiments, revised our paper, highlighted the changes, and tried our best to address all the comments below. We hope the responses and revisions will be viewed favorably.
> 1. Experimental results are not clear and convincing. Lottery ticket pruning does as good or better than proposed method in many plots. Do you do hyper-parameter search for different methods? A fair comparison should include different fraction values for lottery style pruning. Similarly when different p/q values are selected one probably needs to pick different hyper-parameters. Also, plots needs to be improved to highlight the differences between methods. Using log on the y axis (Figure-2/3) can help with this I think.
> > Thank you for bringing this point to us. Our experimental results demonstrate that it is possible for our method with proper hyper-parameters to outperform Lottery Ticket pruning and prevent model collapse. We have changed our statement in the revision. Comparing our method with Lottery Ticket pruning, the only difference is that we determine the number of pruned parameters at each pruning iteration with the proposed PQ Index instead of a fixed ratio. We do not explicitly conduct the hyper-parameters search in our experiments. We conduct extensive ablation studies to demonstrate the effect and insights of the hyper-parameters. To compare the performance between our method and baseline methods, we demonstrate Performance vs. Percent of remaining weights in Figure 4(a) and Appendix D.3. In Figure 4(a) and Appendix D.3, our method outperforms Lottery Ticket pruning in the case of neuron-wise and layer-wise pruning. This is because Lottery Ticket pruning prunes each neuron and layer according to a fixed ratio. It may accidentally prune model parameters with a large magnitude. On the contrary, our method can prevent pruning dense neurons and layers and thus preserve the model parameters with a large magnitude, even if the pruning iterations continue.
>
> 2. "sota algorithms such as LT-based pruning" This statement is not correct. See [1], which show rewinding the weights is a bad choice and [2] which shows gradual magnitude pruning often achieves best results. It's fine not to have SOTA results/algorithms but this statement is not correct and needs to be changed. I recommend using another pruning algorithm like gradual magnitude pruning with or without learning rate recycling (see [2]).
> > Thanks for pointing out this literature. We have changed our statements in the revision. We developed our algorithm based on rewinding the weights (Lottery Ticket pruning) because we discovered the interesting hypothesis illustrated in Figure 1. It is also interesting to study if our hypothesis also persists for gradual magnitude pruning. We describe this future work in Appendix A of the revision.
>
> 3. Authors say in the introduction "the sparsity will first decrease when a large model is being effectively regularized", however I haven't seen any evidence for this claim. Have I missed a plot? It would be nice to show this in a regular training run (like resnet in cifar-10 with learning rate decay and 200 epochs of training). Similarly one could think of experiment that show the usefulness of the PQ-Index at predicting pruning rates. Can PQ-Index predict maximum pruning rate over the course of the training? How well this compare to using a fix magnitude threshold?
> > The declining trend of the PQ Index (PQI) at the initial stage of pruning can be seen from the third column of Figure 2(a), 3(a), 5, and 6, and Appendix D. For example, PQI is decreasing at the first several iterations for all methods. In Figure 3(a), we performed experiments with the CIFAR dataset and ResNet18, as you suggested. PQI can be used for determining potential model collapse, illustrated in Figure 1. In particular, the sudden decrease of PQI from a peak in the third column of Figure 2(a), 3(a), 5, and 6 indicates such model collapse. Furthermore, our experiments in Figure 4(a) (neuron- and layer-wise) show that our proposed algorithm based on PQI can prune a network to a desirable sparsity level that remains reasonable accuracy with fewer iterations than using a fixed pruning ratio. Moreover, a small pruning ratio obtained from PQI indicates little room for further pruning. It may serve as a stop criterion for the iterative pruning procedure.
>
> 4. Looking at Theorem-2, $n_r=0$ seems to imply removing only zeros. Is that accurate? If so, would be nice to make this clear.
> > Not exactly. The Inequality (2) in Theorem 2 gives a lower bound for the number of indices $r$ need to retain in a vector. Thus, the retaining ratio obtained by this lower bound is conservative, and we may remove non-zero weights when $\eta_r=0$.

---

> > ### Author Response · Authors · 2022-11-19
> > **Response Continue**
> >
> > 5. Do you include pruned connections (zeros) in your PQI calculations?
> > > No, we did not. We expect to use PQI to reflect the sparsity of networks in the pruning procedure. If we calculate PQI with pruned zeros, the network will become sparser and sparser, and PQI will decrease monotonously.
> >
> > 6. Figure:5a there is a peak on the PQ-Index at T=20. Why is this? With a high score like this I would expect performance not to drop after pruning.
> > > That corresponds to our hypothesis: Initially, PQI decreases due to regularization (the network becomes denser than before, and performance does not decrease). Then PQI starts to increase due to compression (the network becomes sparser than before and performance slightly drops). Finally, PQI suddenly drops from a peak (the network becomes dense again, and the performance significantly drops). We want to point out that a single PQI value does not imply the performance of a network, but the dynamics of PQI during the pruning procedure is a possible indicator.
> >
> > 7. When p=0.5 and q=1, sparsity would be equal to PQ-metric if I understand correctly. d=100, one-hot, sparsity=PQ-index=0.99. It would be nice to do experiments on pretrained models at different widths and sparsities (this can be achieved with different regularization coefficients) and show how well PQ-index provide maximum pruning rate without significant performance drop. It is very difficult to make conclusion in the LT-rewinding setting since, pruning is repeated multiple times and it is not clear a decision was good or not.
> > > $p=0.5$ and $q=1$ is one particular case of PQI. Indeed, when $p=0.5$ and $q=1$, $d=100$, one-hot, PQI=$1-100^{-1}$. We did not use regularization coefficients for compression. In the LT-rewinding case, PQI at each pruning iteration is computed with the remaining model parameters (not including pruned zeros from previous pruning iterations). Our results in Figure 2-4 demonstrate that PQI can provide insights into the maximum pruning rate in the LT-rewinding case.
> >
> > 8. (Page 8: "It aligns with the intuition..") [2] shows that early layers are often better kept dense.
> > > Thank you for bringing this paper to us. We have cited it to support our claim better now.
> >
> > 9. It would be nice to compare the final sparsity distributions found by PQI with the ones provided by the ERK distribution [3].
> > > Thank you for bringing this paper to us. We have cited it in Appendix A.
> >
> > 10. "pruning is the most popular and effective approach" probably quantization had much more impact on this due to the int8 support on many recent hardware.
> > > Thank you for pointing it out. We have changed the statement accordingly.
> >
> > 11. Remark 3 "... does not satisfy the six properties" -> which of the six? I think some of them are satisfied. It would be nice to be more specific here.
> > > Thank you for this helpful comment. We have specified it in Remark 3.
> >
> > 12. Lottery ticket prunes p\% of the remaining weights. So the weights pruned at each iteration decreases over time. Figure:3, it is not clear which methods does better (like many others).
> > > In Figure 2 and 3, we focus on demonstrating the relationship between the performance of iterative pruning and the dynamics of PQI. To better illustrate the difference of performance among various pruning methods, we provide Figure 4 and Appendix D.3. (Performance vs. Percent of remaining weights). In Figure 4(a) and Appendix D.3, our method outperforms Lottery Ticket pruning in the case of neuron-wise and layer-wise pruning.

---

> ### Author Response · Authors · 2022-12-08
> **A kind reminder**
>
> Dear Reviewer dVA5,
>
> We would like to thank you again for the time you dedicated to reviewing our paper and your valuable comments. We believe that we have addressed your concerns. Since the end of Discussion Stage 2 is approaching and we have not heard back from you yet, we would appreciate if you kindly let us know of any other concerns you may have, and if we can be of any further assistance in clarifying any issues.
>
> Best wishes,
>
> Authors

---

### Decision · Program_Chairs · 2023-01-20

**Decision:**

Accept: poster

**Justification For Why Not Higher Score:**

Please see the weakness section of the meta review.

**Justification For Why Not Lower Score:**

see strength section

**Metareview: Summary, Strengths And Weaknesses:**

This paper introduces a norm based metric (PQ-Index) to quantify weight sparsity of a neural network. The metric is designed to satisfy the six properties proposed by earlier research in economy similar to the Gini Index. PQ-Index is then used to decide how much a neural network can be pruned at a given time during the training. Authors compare their method to Lottery-ticket style iterative pruning on CIFAR and Fashion-MNIST.

Strengths:
- Proposal of a grounded sparsity metric, which is novel and interesting
- The scope of ablation and results presented analyze the given method well.
- The authors proposed the sparsity-informed adaptive pruning method based on the PQ index.

Weakness:
- the practical implementation of the method is cumbersome due to additional hyperparameters.
- It's unclear how SAP can reach a pre-defined pruning rate given some iterations since SAP is an adaptive pruning algorithm. Many hyperparameters are related to the pruning rate per iteration-.
-Authors mentioned that "pruning is based on the assumption that parameters with small magnitudes are removable." However, this is not always the case, many methods use other criteria to measure the importance of each weight besides its magnitude.
-The paper didn't include large-scale datasets and models.


Requirements for the camera ready:
Experimental evaluation for proposed pruning method needs to be improved. We recommend authors to focus on showing the utility of PQI directly using the same model but different metrics (e.g. measuring exactly how close it gets to the max-prune rate). Similarly I think the fixed-pruning rate needs to be searched-over for the LT baseline (like trying P=0.2,0.1,0.3) just to make sure gains are not due to better rate average rate (first point in weakness section). Also please switch the log-scale for plots, as it is still very difficult to compare different curves at the moment.



**Note From Pc:**

if the above contains the word "oral" or "spotlight" please see: "oral" presentation means -> notable-top-5% and "spotlight" means -> notable-top-25%. As stated in our emails, we are disassociating presentation type from AC recommendations